# Optimization of a Monobromobimane (MBB) Derivatization and RP-HPLC-FLD Detection Method for Sulfur Species Measurement in Human Serum after Sulfur Inhalation Treatment

**DOI:** 10.3390/antiox11050939

**Published:** 2022-05-10

**Authors:** Barbara Roda, Nan Zhang, Laura Gambari, Brunella Grigolo, Cristina Eller-Vainicher, Luigi Gennari, Alessandro Zappi, Stefano Giordani, Valentina Marassi, Andrea Zattoni, Pierluigi Reschiglian, Francesco Grassi

**Affiliations:** 1Department of Chemistry “G. Ciamician”, University of Bologna, 40126 Bologna, Italy; nan.zhang2@unibo.it (N.Z.); alessandro.zappi4@unibo.it (A.Z.); stefano.giordani7@unibo.it (S.G.); valentina.marassi2@unibo.it (V.M.); andrea.zattoni@unibo.it (A.Z.); pierluigi.reschiglian@unibo.it (P.R.); 2byFlow SRL, 40129 Bologna, Italy; 3Laboratorio RAMSES, IRCCS Istituto Ortopedico Rizzoli, 40136 Bologna, Italy; laura.gambari@ior.it (L.G.); brunella.grigolo@ior.it (B.G.); 4Unit of Endocrinology, Fondazione Istituto di Ricovero e Cura a Carattere Scientifico Ca’ Granda-Ospedale Maggiore Policlinico, 20122 Milan, Italy; cristina.eller@policlinico.mi.it; 5Department of Medicine, Surgery and Neurosciences, University of Siena, 53100 Siena, Italy; luigi.gennari@unisi.it

**Keywords:** hydrogen sulfide pool, biomarkers, bone metabolism, high-performance liquid chromatography with fluorescence, monobromobimane, sulfur species

## Abstract

(1) Background: Hydrogen sulfide (H_2_S) is a widely recognized gasotransmitter, with key roles in physiological and pathological processes. The accurate quantification of H_2_S and reactive sulfur species (RSS) may hold important implications for the diagnosis and prognosis of diseases. However, H_2_S species quantification in biological matrices is still a challenge. Among the sulfide detection methods, monobromobimane (MBB) derivatization coupled with reversed phase high-performance liquid chromatography (RP-HPLC) is one of the most reported. However, it is characterized by a complex preparation and time-consuming process, which may alter the actual H_2_S level; moreover, a quantitative validation has still not been described. (2) Methods: We developed and validated an improved analytical protocol for the MBB RP-HPLC method. MBB concentration, temperature and sample handling were optimized, and the calibration method was validated using leave-one-out cross-validation and tested in a clinical setting. (3) Results: The method shows high sensitivity and allows the quantification of H_2_S species, with a limit of detection of 0.5 µM. Finally, it can be successfully applied in measurements of H_2_S levels in the serum of patients subjected to inhalation with vapors rich in H_2_S. (4) Conclusions: These data demonstrate that the proposed method is precise and reliable for measuring H_2_S species in biological matrices and can be used to provide key insights into the etiopathogenesis of several diseases and sulfur-based treatments.

## 1. Introduction

Hydrogen sulfide (H_2_S) is a gasotrasmitter that plays important physiological roles as a vasorelaxant [1], a neuromodulator [2,3], a regulator of renal physiology [4] and the endocrine system [5,6], a modulator of gastrointestinal mobility [7,8], and as an inhibitor of cancer cell growth [9]. Moreover, H_2_S is involved in the regulation of bone cell differentiation and was shown to play an anabolic role in various bone wasting conditions. Endogenous H_2_S is synthesized and degraded by mammalian tissues at relatively high rates [10] and is detectable in blood circulation [11]. It is produced in mammalian cells as a byproduct of the enzymatic reaction catalyzed by cystathionine γ-lyase (CSE), cystathionine β-synthase (CBS), and 3-mercaptopyruvate sulfurtransferase (3MST), through the transsulfuration pathway [12]. In addition to this enzymatic pathway of H_2_S generation, a non-enzymatic mechanism for H_2_S release from sulfur-containing amino acids (SAA), catalyzed by iron and Vitamin B6, has been recently characterized and may contribute to sulfide homeostasis under certain physiological conditions [13].

H_2_S is slightly soluble in aqueous solution and can exist as different species. It acts as a weak acid with two acid dissociation constants, pKa_1_ of 7.4 and pKa_2_ of 19.0. The stability of sulfide is greatest under acidic conditions, and it is present as volatile H_2_S. However, at higher pH, H_2_S exists primarily as HS^−^; while gaseous H_2_S and sulfide anion S^2-^ are present, respectively, at low and at negligible concentrations. As a consequence, as the physiological pH level in blood is around 7.4, H_2_S is primarily found in the form of HS^−^. [14]. Moreover, H_2_S bioavailability is regulated through its conversion into different chemical forms, or pools, due to the strong reducing potential of H_2_S and its affinity for thiol groups in proteins [14,15,16]. Acid-labile sulfide and bound sulfane sulfur pools are the two main stored biochemical sources, which act as sulfide buffering regulators by releasing H_2_S via different chemical reactions. Acid-labile sulfide consists of iron-sulfur clusters (Fe-S) contained in iron-sulfur proteins (ferredoxin, glutaredoxin), which can release H_2_S below a pH of 5.4. Bound sulfane sulfur exists as compounds containing sulfur-bonded sulfur, including persulfides (RS-SH), hydropolysulfides (RS_n_-SH), polysulfide (RS-S_n_-SR), thiosulfate (S_2_O_3_^2−^), thiosulfonates (RSO_2_SR’), polythionates (S_n_(SO_3_)_2_^2−^), and elemental sulfur S^0^ [17] and peptide-protein bound (e.g., haemoglobin, myoglobin, neuroglobin); and can release H_2_S under reducing conditions. Notably, all these sulfur species have been established to trigger sulfide-dependent biological events [18,19,20].

The mounting evidence of an important role of the H_2_S system in preclinical studies and in therapeutic applications has stimulated investigation of the correlations between H_2_S levels and the onset and the prognosis of certain diseases, including SARS-CoV -19, diabetes, cardiovascular diseases, osteoporosis. Thus, H_2_S species quantification in biological matrices remains a critical challenge in medicine.

The quantification of H_2_S species in samples encounters several difficulties due to the gaseous nature of the molecule, particularly its volatility, redox lability, and most importantly, its low steady-state concentration [21]. Depending on the chemistries of the methods used, marked differences in H_2_S levels in physiological fluids, spanning from nanomolar to hundreds of micromolar, have been reported in the literature [17,22,23]. Moreover, the methods most often employed for H_2_S measurements are associated with substantial artifacts [24] and various complications can arise, depending on the samples analyzed, e.g., quantitation of steady-state H_2_S levels in tissues [14]. The methods include colorimetric analysis monitoring methylene blue formation, use of a sulfide ion-selective or a polarographic electrode, gas chromatography (GC) with flame photometric or sulfur chemiluminescence detection [25,26], ion chromatography, high performance liquid chromatography with fluorescence detection (HPLC-FLD) analysis of the monobromobimane (MBB) derivative of sulfide [22,27,28], and the use of sulfide-sensitive fluorescent dyes [29,30]; an isomer of MBB was also shown to have high sensitivity for the quantification of H_2_S in blood [31]. Among the new methods for the detection of sulfur biological pools the use of green-fluorescent-protein (GFP)-based probes [32] and a resonance synchronous spectroscopy-based method (RS2) [33] have been reported. Notably, GFP-probes can detect real-time polysulfides levels in live cells and subcellular organelles, with minimal interference due to reactive oxygen species (ROS) scavenging, while the RS2 method can detect intracellular polysulfides and persulfides by comparison of species-specific RS2 spectra and intensities at physiological pH.

HPLC-FLD based on MBB derivatization presents interesting possibilities for the analytical determination of H_2_S species levels, due to its high sensitivity and selectivity, with low nanomolar limit of detection (LoD) [28]. MBB is a fluorescent cell-permeable alkylating agent that reacts with extra- and intracellular sulfide pools and sulfhydryl-containing biomolecules. It is used as an alkylating agent and rapidly derivatized H_2_S under gentle conditions [34]. The product of this reaction, sulfide dibimane (SDB), is fluorescent and stable and can be separated and quantitated for an accurate determination of absolute H_2_S species levels in various biological media [35,36].

However, several experimental analytical conditions of the MBB protocol derivatization, especially in the first step of alkylation of H_2_S, can influence endogenous balance of H_2_S species in biological samples, possibly leading to readouts that do not represent the true cellular speciation and correct “free sulfide levels”. In particular, alkylation has been shown to perturb sulfur speciation and influence sulfide detection in a concentration- and time-dependent manner: at high concentration MBB can cleave longer dialkyl polysulfide chains and extract H_2_S from these bound sulfane sulfur pools, thus, shifting speciation of sulfur species [37]; MBB can liberate sulfide from sulfide pools when increasing reaction times above 7–10 min [37].

Moreover, light exposure can influence the measurement, given that MBB is a light sensitive reagent [35]; fluorescent light and sunlight cause a significant loss in measured sulfide levels [28]. Another critical experimental setting is the temperature of reaction: while 4 °C was shown to minimize enzymatic production or degradation of H_2_S [35], raising the temperature to 37 °C increased the values by 4–5 times compared to room temperature (RT) [28]. Finally, other critical parameters include pH (increased pH results in increased release [28]); the influence of O_2_ (1% represents the ideal condition for derivatization yield) [22]; and the presence of chelators (DTPA, EDTA increased derivatization yield) [28]; while the tubes used for blood sampling with or without additive can interfere with the reaction yield [28]. In addition, a validated calibration procedure has not yet been described.

Most papers have described protocols for free H_2_S quantification in human serum; however, only a very few papers have reported the application of the HPLC-FLD method for H_2_S species (free, acid-labile, and bound sulfane) determination [27,35].

Based on the MBB methods proposed, here we optimized and validated a highly sensitive, robust, and high throughput HPLC-FLD method for the selective determination and quantification of free, acid labile, and bound sulfane sulfur in human serum samples. Among the derivatization parameters, temperature and sample handling conditions were more thoroughly studied to develop a robust protocol able to limit potential alterations of H_2_S pool. A validation of the calibration procedure was proposed for the first time; linearity, LoD, and reproducibility were determined.

Moreover, to assess the efficacy of the optimized method in a clinical setting, we analyzed the amount of serum H_2_S species in patients subjected to an external source of H_2_S, such as the sulfurous vapors of a local thermal spring; this model allowed us to perform a perspective measurement of H_2_S species in human serum samples, thereby tracking the change of the different H_2_S species upon exposure to a relevant environmental source of H_2_S.

## 2. Materials and Methods

### 2.1. Chemicals and Supplies

Anhydrous sodium sulfide (Na_2_S, Sigma-Aldrich, Cat. No. 407410-10G HPLC grade, purity ≥98.0%, product of USA), Monobromobimane (MBB, Sigma-Aldrich, Cat. No. 4380-10MG purity ≥97%, product of USA), Tris(hydroxymethyl)aminomethane (Sigma-Aldrich CHEMIE GmbH, Cat.:15,456-3 Lot.:01927DE-417 Tris base, purity ≥99.9%, ultrapure grade, product of Germany), Trifluoroacetic acid (TFA, Sigma-Aldrich, Cat. No. T6508-500MLHPLC grade, purity ≥99.5%, product of USA), 5-sulfosalicylic acid dihydrate (SSA, Sigma-Aldrich CHEMIE GmbH, Cat. No. S2130-100G purity ≥99.0%, product of Germany), Sodium dihydrogen phosphate dihydrate (NaH_2_PO_4_·2H_2_O, Sigma-Aldrich, Cat. No. 71500-250G, product of Italy), Diethylenetriaminepentaacetic acid (DTPA) (purity ≥99.0%, Sigma-Aldrich, Cat. No. D6518-5G, product of USA), Tris(2-carboxyethyl) phosphine (TCEP, Sigma-Aldrich, Cat. No. C4706-2G), purity ≥98.0%, product of India), bovine serum albumin lyophilized powder (BSA, Sigma-Aldrich, Cat. No. A2153), Hydrochloric acid (HCl, 37%, Sigma-Aldrich CHEMIE GmbH, Cat. No. 30721-1L-M, product of Germany), and Phosphoric acid (H_3_PO_4_ 85%, Sigma-Aldrich, Cat. No. 695017-500ML) were used to adjust the pH value of the buffer solution and purchased from Merck (Darmstadt, Germany). Acetonitrile (ACN, Sigma-Aldrich, Cat. No. 34851-2.5L, purity ≥99.9%, ultrapure grade, product of France) and LC-grade methanol (MeOH, Sigma-Aldrich, Cat. No. 34860-1L-R) were obtained from Merck.

The purified deionized water used throughout the study was obtained from a Milli-Q purification system (ELGA LC134, 0.2-micron filter, 18.5 mΩ cm^−1^, Merck Millipore, Darmstadt, Germany).

Sterile 2 mL BD Vacutainer tubes with clot/activator (BD Vacutainer^®^, Plastic Serum Tube 2 mL with Red Hemogard Closure. Cat. no. 368493, additive: Silica (Clot Activator)) were used to collect serum samples from patients.

BD empty tubes (Vacumed^®^, 13 × 75 mm no additive x 3 mL of blood, Cat. no. 42912, white cap) were used during the determination of different H_2_S levels in serum. BD Quincke point spinal needles 20G 0.9 × 90 Mm were used during the speciation protocol.

### 2.2. Preparation of Buffer Solutions and Reagents

The 100 mM phosphate buffer solution (PB-A) was prepared by mixing 80 mM NaH_2_PO_4_·2H_2_O with DTPA at a final concentration of 0.1 mM; pH value was adjusted to 2.6 by adding 0.1M phosphoric acid.

The 100 mM phosphate buffer solution (PB-B) was obtained by addition to PB-A solution TCEP at a final concentration of 1.0 mM.

The 100 mM Tris-HCl buffer solution was prepared by mixing Tris base with DTPA at a final concentration of 0.1 mM. Then the pH value was adjusted to 9.5 by adding 0.2 M hydrochloric acid.

A 1.5 mM solution of MBB was prepared in acetonitrile. This solution was kept in an amber container and protected from light to avoid photolysis.

### 2.3. Standard and Solutions

Sodium sulfide (Na_2_S) was employed as a source of H_2_S for standard solutions. A 5 mM stock solution of Na_2_S in water was freshly prepared and stored in an opaque centrifuge tube at RT. Seven calibration standards (0.8, 1.6, 3, 6, 12.5, 25, and 50 µM Na_2_S) were then prepared by diluting the original stock solution with water.

Water and solvents were deoxygenated by sonication before usage, and all working standard solutions were freshly prepared for derivatization every day.

### 2.4. Patients and Serum Samples

The validated procedure was applied to serum samples obtained from 4 post-menopausal women (age: 55 ± 2.9) recruited at the Rizzoli Orthopedic Institute. Following the signing of the informed consent (AVEC 442/2018/OSS/IOR), the patients were subjected to inhalation treatment of sulfurous waters at the thermal spring of Castel San Pietro Terme (Bologna, Italy), where the concentration of H_2_S is 14.6 mg/L.

In detail, each patient underwent a cycle of 30-min inhalation treatment for 12 consecutive days. Blood samples were taken before the treatment (T0), immediately after the 12-day treatment (T1) and three days after completion of the treatment (T2). Blood samples were collected in “serum collection tubes”, tubes were carefully sealed to avoid leaking of the gaseous phase, and the serum was collected by centrifugation of blood at 3500 rpm for 15 min at 4 °C. Serum samples were transferred in polypropylene tubes and immediately frozen at −20 °C, for up to 1 month until analysis, with no further cycles of freezing and thawing.

### 2.5. Derivatization Procedure

Before the derivatization procedure, all the materials were placed in a hypoxic chamber (BENCHTOP GLOVE-BOX W/GAZ PORT, Fisher Scientific Rodano, Italy), which was then purged with nitrogen gas to 1% O_2_. All working solutions were freshly prepared, and the entire derivatization procedure was carried out as quickly as possible and under light.

The protocol chosen for the derivatization of H_2_S with MBB was the following:

First, 30 μL of standard solution or blank or serum sample was mixed with 70 μL Tris-HCl buffer solution and 50 μL of 1.5 mM MBB. Eppendorf PCR tubes containing these reagents were immediately capped, vigorously vortexed for 5 s, and incubated in the dark for 30 min at RT. Then, the reaction was stopped by adding 50 μL of 200 mM 5-sulfosalicylic acid to tubes, which were further vortexed for 5 s. Finally, 200 µL of the resulting solution was transferred from PCR tubes to autosampler vials equipped with a 200 μL plastic insert vial for the HPLC quantification of SDB derivatization product.

### 2.6. Detection of H_2_S Species in Serum Samples

To obtain free H_2_S, HS−, S^2−^ (‘**free**’ *H_2_S levels*, the derivatization procedure was applied to serum sample by centrifuging the final reaction samples at 13,000× *g* for 10 min before transferring supernatants into HPLC vials and analyzing them with HPLC-FLD.

The acid-labile sulfide and bound sulfane sulfur were detected following the speciation protocol based on the selective liberation of H_2_S, as already described [35]. Briefly, a volume of 50 μL of serum samples was added into two sterile empty BD vacutainer collection tubes without additive. Subsequently, 450 μL of PB-A or PB-B was added to the two tubes, respectively. Notably, PB-A solution, which was maintained at a pH 2.6, acidificated the sample and released H_2_S from acid labile pools; PB-B contains TCEP, which cleaves the disulfide bonds of sulfane sulfur and releases sulfane sulfur atom. After 30 min incubation on a rocker, all tubes were placed in a hypoxic chamber and the solution was removed using a syringe with a spinal needle (1 mL) without inverting the tubes. Hereafter, 500 μL of Tris-HCl buffer solution was added, to trap the volatilized H_2_S for 30 min incubation on the rocker. All procedures of volatilization and trapping of H_2_S were conducted in a hypoxic chamber at RT.

Finally, an aliquot of solution was derivatized from each tube following the procedure described above in paragraph 2.5 and analyzed using HPLC-FLD. The sample treated with the acid liberation protocol (PB-A) gave the acid sulfide value (***acid H_2_S levels***); while the sample treated in acid condition, with the addition of the reducing agent TCEP (PB-B) for disulfide bonds, gave the total sulfide value (***total H_2_S levels***). Finally, the acid-labile sulfide level and the bound sulfane sulfur levels were calculated as follows:***acid-labile sulfide level*** = ***acid H_2_S levels − ‘free H_2_S’ levels***
***bound sulfane sulfur level*** = ***total H_2_S levels − acid H_2_Slevels***

### 2.7. Instrumentation and Analytical Methods

Derivatized samples were separated and SDB was quantified using an Agilent 1260 Infinity HPLC system, equipped with a G1379B degasser, G1312B binary gradient pump, G1329B autosampler, G4212B diode array detector, and G1321A fluorescence detector; and a Chemstation Chromatography Workstation. Separations were carried out at RT on an Agilent Eclipse XDB-C18 column (4.6 × 250 mm), with an average particle size of 5.0 µm.

Analyte separation was performed using a binary mixture comprising a mobile phase A (water) and a mobile phase B (acetonitrile), which were adjusted with 0.1% (*v*/*v*) trifluoroacetic acid (TFA), at a flow rate of 0.6 mL/min.

The gradient elution started at 85:15 (*v*/*v*) and decreased to 68% water in 3 min. Then, it remains the gradient downward trend drop to 55% water for 13 min. Afterwards, the system was kept in isocratic elution mode for 1 min and then brought back to initial conditions 85:15 (*v*/*v*) in 3 min and left in this condition for 3 min to stabilize the pressure of the chromatographic system; thus, completing the chromatographic separation in 23 min.

The excitation and emission wavelengths of the fluorescence detector were set to 390 and 475 nm, respectively. Quantitative determinations were carried out using peak area measurements at the emission wavelength.

The sample volume injected in the chromatographic system was optimized and set at 30 µL. All solutions were filtered prior to analysis through a 0.2 μm syringe filter and injected in three replicates. The data were integrated using an automated software system. Chromatographic peaks were checked, and identification was achieved by comparing retention times.

To confirm the peak identity, an UPLC chromatographic system model (ACQUITY H-CLASS) coupled with a model Xevo G2-XS QTof mass spectrometer (UPLC-QTOF-MS, Waters Corp. Milford, MA, USA) was used. The mobile phase, gradient program, and column were the same as used for the HPLC-FLD system. The injection volume was 10 μL.

### 2.8. Method Validation and Statistical Analysis

The calibration procedure was defined and validation parameters, such as linearity, LoD, limit of quantification (LoQ), intra- and inter-day precision, and matrix effect, were determined.

Calibration curves were performed using seven Na_2_S standards (0.8—50 µmol/mL), as described in paragraph 2.3. The absolute peak area was plotted against the different derivatization product concentrations, and the curves were fitted, both by polynomial and linear regression analysis.

LoD and LoQ were calculated for the linear model as, respectively, 3.3 and 6 times the ratio between the root mean squared error (RMSE) and the slope of the model. RMSE is defined as the mean of the squared differences between the experimental responses and the response values recalculated by the model.

Five replicates of each point were analyzed, to determine the intra- and inter-day precision. This process was repeated three times over three days, to determine the inter-day precision, using freshly prepared calibration curves. Significant differences between inter- and intra-day replicates were checked by Student’s *t*-test.

All values are reported as mean ± standard deviation. Statistics were calculated using R statistical software (R Core Team, Vienna, Austria).

Data obtained from human samples were analyzed with an ANOVA test for repeated measures, followed by a Dunnett multiple comparison test.

## 3. Results

The aim of the presented work was the optimization of the experimental procedure for the MBB HPLC-FLD method and its validation for the quantitative determination of H_2_S levels in its multiple forms in human serum samples. Based on the reported papers [28], some crucial parameters were optimized to define a high throughput and robust analytical protocol, to establish reliable bioavailable concentrations of sulfide able to highlight its potential role as a biomarker for diagnostic and therapeutic applications. In addition, a robust quantitative validation was performed. For the optimization procedure, aliquots of a T0 serum sample were used.

### 3.1. RP-HPLC-FLD Separation of Derivatization Product

Separation performance for the SDB product was verified, implementing an already reported HPLC-method [27] and increasing the injected volume to 30 mL. Representative chromatograms of the derivatization protocol applied to a blank sample (water), a standard solution (6 μM Na_2_S), and a serum sample are reported in Figure 1a. The chromatogram of a MBB solution diluted by acetonitrile is also shown, to indicate the retention time of MBB excess. A standard sample and blank sample were used to attribute the SDB peak in serum samples. SDB, highlighted with a red dotted square, was eluted at t_R_ = 11.8 min, while the excess of MBB was eluted later (t_R_ = 12.8 min).

The application of the UPLC-QTOF-MS analysis allowed the SDB peak identity to be confirmed. Figure 1b shows the MS-spectrum of the SDB peak. The m/z of 415.14 corresponds to the molecular ion of SDB [38].

### 3.2. Sulfide Levels Quantification, Method Development

#### 3.2.1. Derivatization Method Optimization

The first step of the derivatization procedure is a bimolecular reaction between H_2_S and MBB, to give the intermediate sulfide monobimane species; then, a nucleophilic attack with a second MBB molecule yields the SDB product. However, the conditions used to optimize the yield of the reaction may strongly influence the biological equilibria of H_2_S levels, affecting the ratio between the different forms of sulfide. A compromise between high yield and preservation of equilibrium, together with the analytical performance of the method, must be established. Based on previous studies on the HPLC-FLD-MBB method for H_2_S species quantification in serum samples [22,35], we applied pH 9.5, Tris HCl as buffer, 1% O_2_, a 30 min reaction time, and a dark environment as reaction conditions; moreover, we considered some crucial parameters such as MBB concentration, temperature, and sample handling conditions for further optimization.

##### MBB Concentration

First, we tested the effect of lowering the concentration of MBB from 10 mM, the one most used in the literature [22,27,35], to 1.5 or 0.15 mM for the derivatization of a 12.5 μM Na_2_S solution. The resulting chromatograms are shown in Figure 2. Use of 10 mM MBB provided the highest peak area for SDB, but also two high peaks corresponding to by-products. The use of the lowest MBB concentration (0.15 mM) reduced the formation of by-products but also the SDB peak intensity, due to incomplete derivatization of free H_2_S. Notably, the use of 1.5 mM MBB allowed the preservation of SDB peak intensity and reduced the by-product formation, together with a reduced MBB solution consumption.

##### Temperature

We then tested the effect of increasing temperature from RT to 50 °C for the derivatization of a 12.5 μM Na_2_S solution. To clarify the effect of these conditions on the biological samples, a serum sample and a simulated body fluid solution (SBF) were also analyzed. The SBF solution was prepared by dissolving 4 mg/mL serum albumin in phosphate buffer saline pH = 7.4, to mimic the composition of human serum, in terms of ion and protein concentrations.

The peak area values obtained for SDB are reported in Table 1.

Increasing the temperature caused an increase in the SDB formation for all the analyzed solutions, as shown in Table 1. Both in the standard solutions and serum sample, a nearly 3-fold increase in the reaction yield was observed with increasing temperature. Notably, SDB was detected in SBF only at 50 °C; thus, suggesting the extraction of H_2_S from proteins contained in SBF. The release of HS^-^ within the sample would cause an increase in the signal associated with SDB, thus altering the levels of the various types of endogenous sulfide detected by this method. Accordingly, RT was chosen as the condition for the optimized derivatization procedure.

##### Serum Handling (Storage Conditions, Aging, Dilution)

The effect of serum handling (storage conditions, aging and dilution) during the pre-analytical phases was considered.

As for the sample storage procedure, no significant differences were found between the area of the SDB peak obtained from the three aliquots of serum sample stored at −20 °C for up to 3 months (Figure 3a). Conversely, repeated freeze–thawing cycles lead to a significant reduction of the SDB signal obtained from the same serum sample (Figure 3b).

Due to the complexity of the serum matrix, and to reduce potential chemical and instrumental interferences, a 1:1 sample dilution step with physiological solution was considered. The derivatization yield for free H_2_S was compared for diluted and undiluted samples (Figure 3c).

However, the diluted sample showed a higher yield for the SDB formation, indicating that potential equilibrium shifts had occurred after the dilution step. Therefore, in the optimized protocol, aliquots of serum samples (at different times of storage at −20 °C, but only thawed once) were directly analyzed following the derivatization procedure described, without any dilution in a physiologic solution.

##### Speciation Protocol Tubes

For the acid-labile sulfide and bound sulfane sulfur quantification, the protocol described in part 2.6 was applied. A 2-mL BD vacutainer tube with clot-activator/gel was initially used during sample treatment for H_2_S pool release. However, when the derivatized Tris-Cl (blank) and serum samples were analyzed, even the blank sample showed a peak at the retention time of SDB (t_R_ = 11.8 min) with an area equivalent to a 15 μM Na_2_S standard solution (data not shown). The identity of this peak was confirmed through UPLC-MS analysis, and the MS Intensity of SDB peaks from the analyzed samples (acid H_2_S from serum, total H_2_S from serum, acid H_2_S from Tris-Cl and total H_2_S from Tris-Cl) are reported in Figure 4.

To further investigate the origin of this peak in the blank samples and potential interferences due to the reagents and sample treatment, several blank samples were derivatized for the free sulfide determination using the 2-mL BD vacutainer tube with clot-activator/gel instead of the PCR tube, and the resulting products were analyzed with UPLC-QTOF-MS. All the analyzed samples showed the peak at t_R_ = 11.8 min, and the identity of SDB was confirmed with the MS-spectra (Figure 4). Despite the serum sample showing higher values for H_2_S species, significant signals were also obtained in the Tris/PB/H_2_O used as blank samples for the free determination. The results indicate potential interferences due to the silicone adsorbed on the 2-mL BD tubes used for all derivatizations. Indeed, when empty BD tubes without additive were used for the speciation protocol, no FLD signal was detected at the derivatization time for SDB with blank samples (t_R_ = 11.8 min); data not shown. Accordingly, BD tubes without additive were used in the optimized protocol.

#### 3.2.2. Calibration Curve: Optimization and Validation

Na_2_S solutions in the concentration range 0.8–50 µM were prepared, derivatized, and used to determine the calibration curve. We decided to explore a wide range of H_2_S concentrations, based on the levels of H_2_S in serum samples reported in literature [28,35], and because we could not predict from the beginning where the H_2_S amounts within serum samples would have fallen. Moreover, given that some authors used spiked serum samples with the addition of Na_2_S as calibration solutions [31], we explored a procedure for standard addition, to limit the matrix effect. We then compared the SDB peak area obtained from Na_2_S standard solutions and from a spiked serum sample with the same standard concentrations. The results showed that the addition of a standard solution of Na_2_S to the serum sample gave a lower value of SDB peak area, with a concentration dependent trend (Table 2). These data indicate that after the addition of Na_2_S to serum sample, a portion of sulfide can be trapped by the matrix, making the correlation between SDB product yield and sulfide concentration unreliable. Therefore, this confirmed the choice of determining the calibration curve for the HPLC-FLD MBB method using Na_2_S standard solutions.

The wide calibration range selected requires the use of a polynomial regression curve, as reported in Figure 5a. The calibration curves were calculated using a second-degree polynomial regression (parabolic), where the independent variable (*x*) is the standard concentration, and the dependent variable (*y*) is the area of the FLD chromatogram. The resulting model equation is $y=a+bx+cx^{2}$. The curve parameters are summarized in Table 3. The R^2^ value of 0.998 and the model p-value close to 0 demonstrate that the parabolic model is a good fit for the data.

To assess the reproducibility of the measures, three replicates of each standard were prepared in different days. Inter-day reproducibility was successfully verified by performing, for each concentration, a Student’s *t*-test between the inter-day replicates and the standards used for the calibration. The relative *p*-values never went under the chosen significance level (α = 0.05), showing no significant differences between the standards and the inter-day replicates.

To further validate the parabolic model, a leave-one-out cross-validation (CV) was performed [39]. This method consists in removing an experimental point from the dataset, computing the model with all the other points, and then projecting the removed point onto the model. This procedure was repeated for all the points, and the recalculated responses (in this case the FLD areas) were compared to the experimental ones. The comparison could be performed by computing a linear regression between the recalculated and experimental responses ($y_{recalculated}=a+by_{experimental}$): the intercept (*a*) and the slope (*b*) of such model should be not significantly different from 0 and 1, respectively, indicating a good match between the recalculated and experimental values. Moreover, the responses can be recalculated by projecting them directly onto the original model, without removing any point, and calculating the same linear regression (Calibration method). The results of such procedure are reported in Figure 5 and Table 4. Figure 5b shows that the two lines (blue and red) coincide with the first quadrant bisector. The line parameters reported in Table 4 show, besides the goodness of the models due to R_2_ being close to 1 and the low *p*-values, that the two intercepts and slopes, also considering the corresponding standard deviations, are not significantly different from the ideal values of 0 and 1.

Indeed, high standard deviations were obtained for the calculated concentrations (data not shown) when the twelve unknown samples (four samples collected at three different times of H_2_S inhalation) were interpolated, making all the calculated concentrations not significantly different from zero. These results revealed the poor prediction ability of the parabolic model. However, we noticed that the interpolated areas of the unknown samples were always in the range 5–15 AU. This means that the unknown samples were interpolated in the lowest concentration range of the curve, far from the centroid (that is around 600 AU). In general, for all regression models, the standard deviation calculated for a projected sample is lowest if the sample is close to the model centroid, while it increases, also dramatically, in the external portions. Therefore, the high standard deviations could be due to the non-optimal performance of the model in that response region.

Therefore, we decided to reduce the standard concentration range to 0.8–6 µM only, corresponding to a range whose centroid is close to the unknown signals. In this way, it was also possible to simplify the regression model to a linear, rather than parabolic, model. The calibration line (in this case in the form y=a+bx) is reported in Figure 6, and the corresponding parameters are reported in Table 5. Both Figure 6 and Table 5 indicate the good fit of the standards to a linear model in the restricted concentration range. R_2_ is close to the ideal value 1 (0.995) and the *p*-value of the model is highly significant (4.77 × 10^−13^). Moreover, the intercept is not significantly different from 0.

LoD and LoQ for the method were calculated from the linear model as 3.3×RMSE⁄Slope and 6×RMSE⁄Slope, respectively. LoD was 0.5 µM, while LoQ was 0.9 µM.

### 3.3. Efficacy of the Method on Human Serum Samples

To evaluate the performance of this method in a clinical setting, we quantified the different sulfide species in the serum of four patients enrolled into an interventional clinical trial at Rizzoli Orthopedic Hospital, based on a cycle of 12 inhalation treatments with sulfurous water with high H_2_S content. The linear model was used to calculate the concentration values.

Figure 7 shows that the thermal treatment had a strong effect on the total concentration of H_2_S, which increased significantly from T0 (before the treatment) to T2 (three days after the completion of the 12-day inhalation treatment) by over 40% (*p* < 0.0001).; while the concentration of free H_2_S remained steady treatment throughout the experiment; consequently, the acid labile and the bound sulfane sulfur fractions were also increased three days after the end of treatments. These results highlighted the specificity of the method, which could detect differences in specific H_2_S levels.

While the biological implications of these findings remain to be elucidated, these data show that an exogenous source of gaseous H_2_S affected the serum levels of the different H_2_S pools.

## 4. Discussion

A wealth of preclinical studies has established that free H_2_S and relative H_2_S species act as important signaling molecules in cells and tissues. Moreover, they have emerged as important determinants of susceptibility or prognosis in certain pathologies and may work as key biomarkers [40,41,42,43].

In the context of musculoskeletal diseases, animal studies have ascertained that H_2_S triggers an antioxidant response sufficient to inhibit osteoclast differentiation [44], and supports osteoblast differentiation and bone formation [45], thereby mitigating the osteoporosis induced by hormone depletion or chronic glucocorticoid treatment [46,47]. Importantly, circulating levels of H_2_S were associated with a lower bone mass in a murine model of post-menopausal osteoporosis [47], and H_2_S biosynthesis was shown to be impaired in the osteoarthritic joint [48]. No information is yet available for regulation of bone tissue by sulfide pools in both physiology and pathology.

The reproducible quantification of H_2_S species (free, acid labile, and bound sulfane sulfur) would help to identify their divergent regulatory roles in several biological processes and help to define the range of concentration of H_2_S released from pharmacological donors and to obtain a targeted delivery of H_2_S at the desired dose. Thus, finding a reliable and sensitive method for H_2_S detection and analysis is still a challenge.

The use of colorimetric detection based on methylene blue has been declining in the literature, due to several limitations: it lacks sensitivity at low (<1 μM) H_2_S concentrations, which makes it inappropriate for measuring lower biological levels of H_2_S; formation of dimers and trimers of methylene blue; interference with other colored substances; and strong acid chemical pretreatment [49]. Electrochemical techniques, such as sulfide ion-selective and polarography, have been developed to detect H_2_S in whole blood and tissues [50]. However, sulfide ion-selective electrodes are prone to fouling and, similarly to the methylene blue method, cannot provide information in real-time or on living tissue and are not sensitive enough for most biological samples [24]. In addition, values obtained from this indirect method tend to be somewhat lower than those using direct methods. Polarography sensors, tend to drift, must be frequently calibrated, and consume sulfide slowly, thus hampering measurements with very small volumes [51]. The most sensitive technique for measuring physiological sulfide levels in pure biological samples, gas chromatography, potentially liberates bound sulfane sulfur because of irreversible sulfide binding or shifts in phase transition equilibria and is not capable of determining the level of H_2_S in real time [52]. Despite the impressive effort being put into the development and validation of accurate and reliable methods for the determination of sulfide levels in the past few years, most of these methods still show several limitations [53].

HPLC-FLD based on MBB derivatization represents the most interesting approach for H_2_S species analysis in plasma or tissue samples, with advantages related to its high sensitivity and specificity.

During sample preparation, biological equilibria can be modified, giving rise to liberation of H_2_S, which may alter the actual levels; consequently, a robust and reproducible analytical protocol must be designed. Complicating the matter is the fact that sulfide exists in multiple forms: free sulfides such as S^2−^, HS^−^, H_2_S, acid-labile and bound sulfane sulfur. These different forms of sulfide make quantitative measurement of bioavailable H_2_S difficult and have led to variable reported levels in the literature.

Here, we applied some reaction conditions (pH 9.5, Tris HCl as buffer, 1% O_2_, 30 min, dark environment) already discussed in previous papers [28and references therein]. The results were in line with these studies and showed no significant differences. Next, we analyzed crucial parameters (MBB concentration, sample handling, tubes for the speciation protocol) that would significantly affect the HPLC-FLD method results.

First, we aimed to reduce the MBB used in the protocol, since high concentration MBB can cleave longer dialkyl polysulfide and this would increase the throughput of the method by consuming less key reagent. The best results were obtained with 1.5 mM MBB, a concentration able to quantify sulfide in a wide interval range, giving greater sensitivity to the technique and reducing the production of by-products. However, previous evidence has shown that a 1.1–10 MBB concentration can extract H_2_S from bound sulfane sulfur pools [37]; therefore, we cannot exclude the possibility that “free sulfide levels” at least partially measured a release from long polysulfides, instead of direct alkylation of free sulfide.

Then, we aimed at optimizing the protocol temperature. The increase in temperature from RT leads to greater yield of the derivatization reaction on Na_2_S standard solutions and serum, with a consequent increase of the sensitivity of the method. However, the SBF used to mimic a biological sample showed a value for SDB formation only at higher temperature; thus, suggesting the extraction of H_2_S from proteins contained in SBF and the alteration of the levels of the various types of endogenous sulfide detected using this method. Albumin contains cysteine residues that are sensitive to thermal degradation, with the consequent release of sulfide in the monoprotonated form (HS^−^). While the Na_2_S standard solution has a reserve of sulfide proportional and limited to the quantity of Na_2_S present, the biological matrix (serum sample) contains various species of sulfide that can be enclosed in metal clusters or linked to proteins, and whose release in solution can be stimulated by external agents; in a selective way, using specific reagent and chemical conditions, as for the speciation protocol; or in a non-specific manner, as by increasing the reaction temperature. Consequently, RT was chosen, to avoid contamination of the endogenous sulfide levels.

Additionally, we tested a dilution of serum samples, but we showed that no dilution should be applied to the serum sample prior to derivatization. In the optimized protocol, 30 µL of serum was directly derivatized without dilution. Since one of the aims of the presented work was to develop a robust protocol able to give reproducible values for H_2_S levels (free, acid-labile, and bound sulfane) in blood, we highlighted that an empty BD-vacutainer must be used during the speciation protocol, to avoid interference in the SDB quantification, due to the presence of clot activator in the tube.

Finally, we studied the standard calibration procedure: we first prepared a stock solution of Na_2_S, then we diluted it to obtain 0.8, 1.6, 3, 6, 12.5, 25, and 50 µM concentrations, which were then derivatized; these standards were used for the calibration curve and a validation was performed with a leave-one-out cross-validation (CV). To our knowledge, this is the first report to employ this validation. Current HPLC-FLD MBB- based protocols for H_2_S levels quantification mostly derivatize the standard solution of Na_2_S and then dilute the relative SDB purified to obtain different calibration standards, to calculate the calibration curve; consequently, issues related to the derivatization procedure, which can strongly affect the reaction yield and quantitative results [22,35,37], are not well considered. Some authors instead reported the use of serum samples spiked with Na_2_S as calibration solutions [31]. In this study, we compared the SDB peak area obtained for Na_2_S standard solutions and a spiked serum sample added at the same standard concentrations. The results indicate that after the addition of Na_2_S to the serum sample, a portion of sulfide can become trapped by the matrix. Therefore, standard addition to the serum matrix does not represent a robust quantitative analytical approach. Thus, the calibration curve was determined by using calibration standards diluted using a stock solution of Na_2_S before derivatization. A LoD of 0.5 μM was determined, indicating a high sensitivity with respect to similar reported methods [27,31].

Then, we tested the present analytical approach in a group of patients undergoing a cycle of inhalation treatment with H_2_S: rich water; this route of administration was chosen because the concentration of free H_2_S in the waters (14.6 mg/L) is biologically relevant, as it falls within the low micromolar concentrations shown to be bioactive in a broad series of preclinical studies. Moreover, whether the exposure to an exogenous source of H_2_S can influence the circulating levels of RSS in humans is an unanswered question, with potential clinical relevance, as the replacement of sulfur was shown to be an effective strategy in certain pathological conditions characterized by lower-than-average RSS in blood. Particularly, we demonstrated that, in a preclinical mouse model of post-menopausal osteoporosis, replacement of decreased H_2_S levels in serum with intraperitoneal administration of sodium hydrosulfide prevented the bone loss occurring due to estrogen deficiency [43]. Notably, the findings of the present study show that this method could detect a change in the different pools of H_2_S in the serum of patients. After the end of treatment (timepoint T2), the total H_2_S increased by over 40% relative to baseline levels, resulting from a similar increase in the acid-labile and bound sulfane sulfur pools; on the other hand, the treatment had no effect on the free H_2_S pool, suggesting that the exogenous free H_2_S quickly reacted with the proteins of the biological matrices in blood. In the context of studies on osteopenia-osteoporosis, this finding could be of relevance, since we could analyze and eventually correlate the H_2_S species modulation in serum, due to sulfurous water inhalation, to biomarkers of bone remodeling.

## 5. Conclusions

The presented work optimized and validated a MBB derivatization method coupled with a HPLC-FLD protocol for H_2_S species quantification in human serum sample and validated for the first time the procedure of calibration. Crucial factors influencing the actual H_2_S species level were excluded from the protocol. Furthermore, we provided evidence of the importance of the procedure of preparation of standard solution and the relative calibration. Overall, the optimized method results in a more efficient derivatization of H_2_S with MBB, with a low perturbation of sample equilibria, giving a robust value for endogenous H_2_S species. Although this method cannot achieve absolute quantification of H_2_S species, it is a good method for relative quantification.

By revealing a modulation in H_2_S species in patients that underwent sulfurous water inhalation, we demonstrated that the method proposed is a reliable tool to measure H_2_S species in biological matrices. This validated detection and quantification method can improve H_2_S species relative quantification in physiology, pathology, and for helping to track H_2_S levels in the context of pharmacological exogenous H_2_S treatments.

## Figures and Tables

**Figure 1 antioxidants-11-00939-f001:**
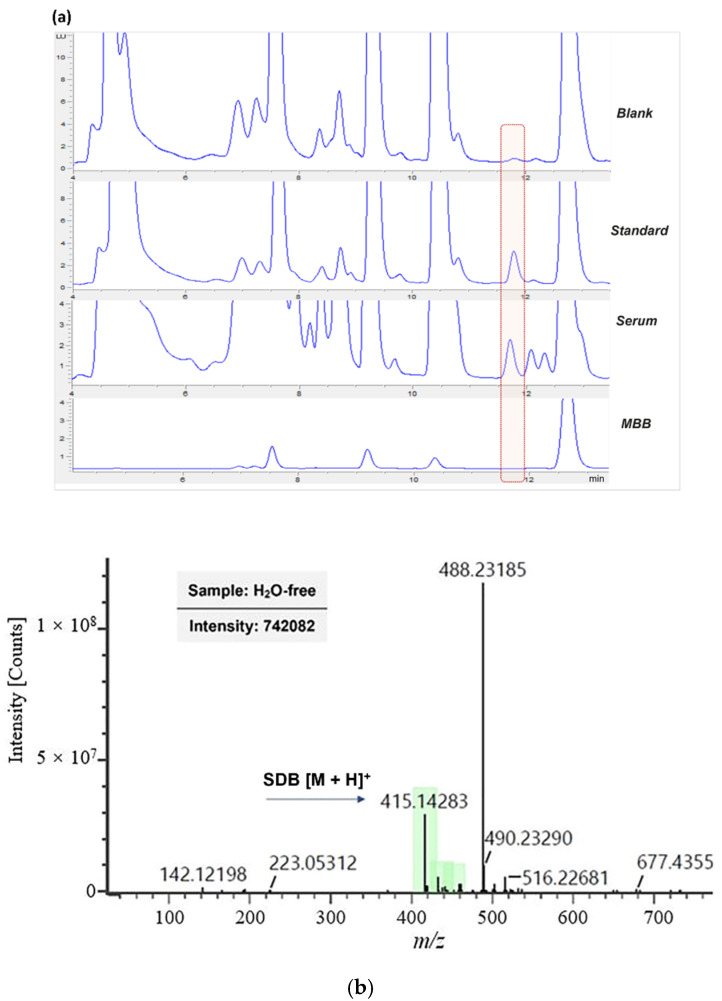
(**a**) Representative HPLC-FLD chromatograms of blank (H_2_O), standard solution (6 μM Na_2_S), serum, and MBB. (**b**) MS spectrum of H_2_S derivatives. The molecular ion of SDB [M + H]^+^ is shown at 415.1 *m*/*z*.

**Figure 2 antioxidants-11-00939-f002:**
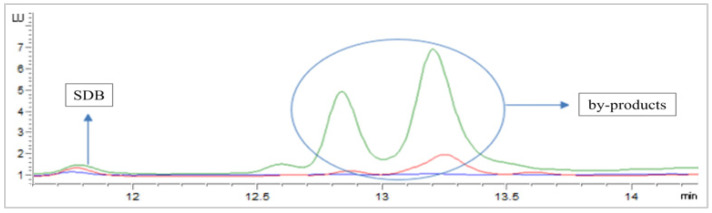
HPLC-FLD chromatograms of 12.5 μM Na_2_S solution derivatized with 10 mM (**green**), 1.5 mM (**red**) and 0.15 mM (**blue**) MBB.

**Figure 3 antioxidants-11-00939-f003:**
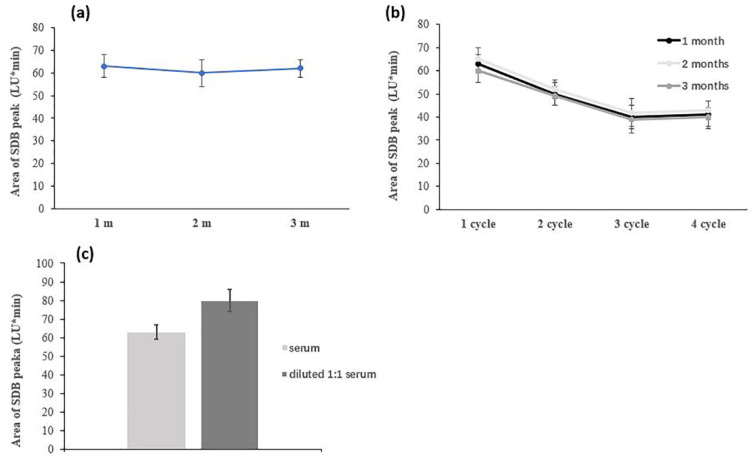
Area of SDB peak for a serum sample (**a**) stored at −20 °C for 1, 2 or 3 months; and (**b**) after repeated freeze–thawing cycles at 1, 2, or 3 months. (**c**) Effect on SDB peak area of 1:1 dilution in physiological solution of serum samples.

**Figure 4 antioxidants-11-00939-f004:**
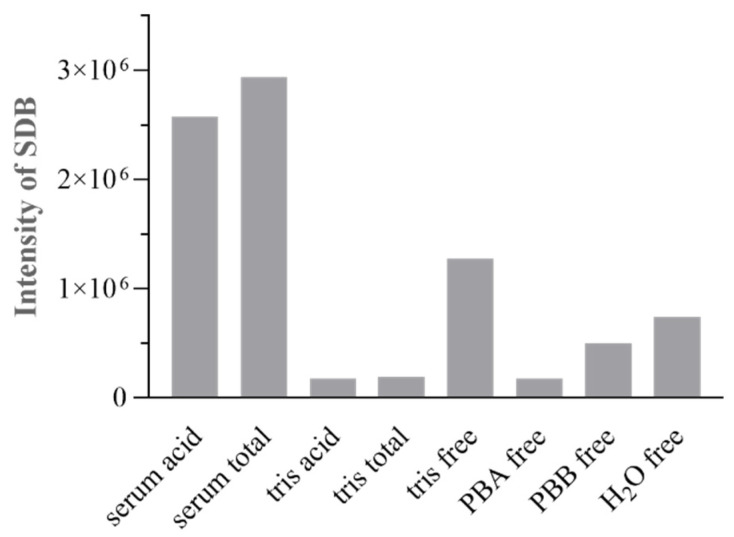
MS intensity of SDB peak for: serum acid (acid H_2_S from serum), serum total (total H_2_S from serum), tris acid (acid H_2_S from Tris-HCl), and tris total (acid H_2_S from Tris-HCl). Tris free (free H_2_S from Tris-HCl), PBA free (free H_2_S from PB-A), PBB free (free H_2_S from PB-B), and H_2_O free (free H_2_S from H_2_O).

**Figure 5 antioxidants-11-00939-f005:**
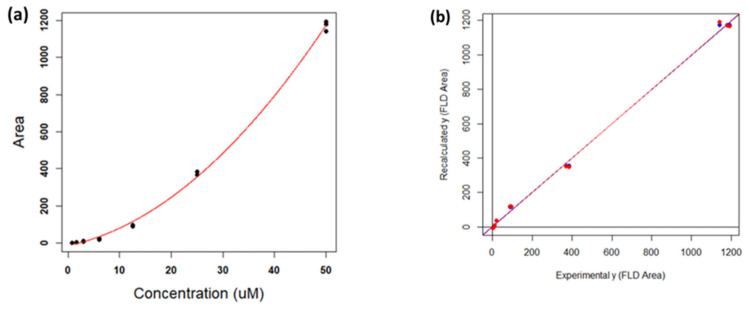
(**a**) Parabolic regression in the concentration range 0.8–50 µM. (**b**) Regression lines of the validation methods using cross-validation (CV) (in red) and calibration (in blue).

**Figure 6 antioxidants-11-00939-f006:**
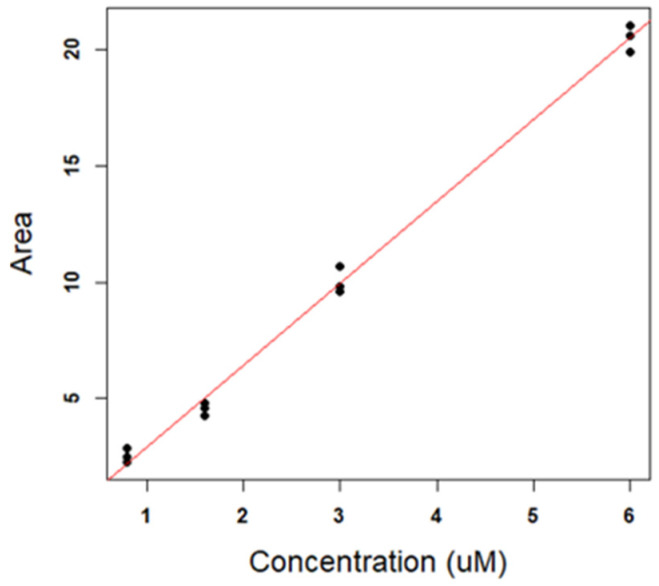
Linear regression in the concentration range 0.8–6 µM.

**Figure 7 antioxidants-11-00939-f007:**
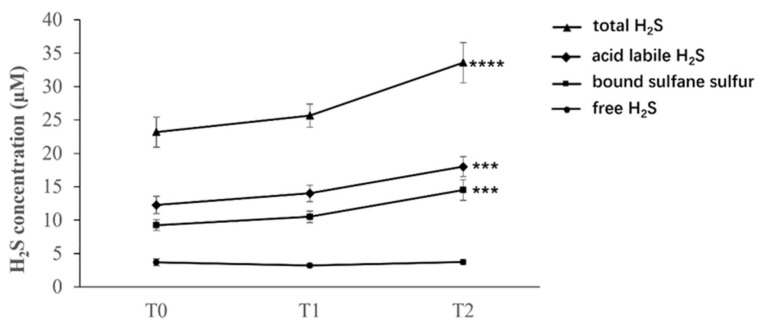
H_2_S levels in serum samples collected before the treatment (T0), immediately after the 12-day treatment (T1), and three days after completion of the treatment (T2). **** = *p* < 0.0001 vs. values at baseline (T0); *** = *p* < 0.005 vs. values at baseline (T0).

**Table 1 antioxidants-11-00939-t001:** SDB peak area of different samples at RT and 50 °C.

SDB Peak Area (LU * min)	RT	50 °C
Na_2_S 12.5 μM	270	800
Serum	63	207
SBF	2	230

**Table 2 antioxidants-11-00939-t002:** SDB peak area of standard Na_2_S sample and spiked serum samples.

	Na_2_S 5 μM/ Serum+ Na_2_S 5 μM	Na_2_S 12.5 μM/ Serum+ Na_2_S 12.5 μM	Na_2_S 25 μM/ Serum+ Na_2_S 25 μM
SDB peak area (LU * min)	25/22	180/100	400/200

**Table 3 antioxidants-11-00939-t003:** Parabolic regression parameters. All values are reported with the corresponding measure unit, AU stands for (FLD) area unit. RMSE stands for root mean squared error.

Parameter	Value
a (AU)	−13.2
s_a_ (AU)	6.61
b (AU µM^−1^)	5.87
s_b_ (AU µM^−1^)	0.930
c (AU µM^−2^)	0.358
s_c_ (AU µM^−2^)	0.0181
R^2^	0.998
RMSE (AU)	18.16
model *p*-value	<2.20 × 10^−16^

**Table 4 antioxidants-11-00939-t004:** CV and calibration regression line parameters.

	CV	Calibration
Intercept	− 0.108	0.429
s_Intercept_	0.685	0.520
Slope	0.999	0.998
s_Slope_	0.0122	0.00969
R^2^	0.997	0.998
RMSE	22.3	17.8
model *p*-value	<2.20 × 10^−16^	<2.20 × 10^−16^

**Table 5 antioxidants-11-00939-t005:** Linear regression parameters. All values are reported with the corresponding measure unit, AU stands for (FLD) area unit.

Parameter	Value
Intercept (AU)	− 0.581
s_Intercept_ (AU)	0.260
Slope (AU µM^−1^)	3.51
s_Slope_ (AU µM^−1^)	0.0749
R^2^	0.995
RMSE (AU)	0.514
model *p*-value	4.77 × 10^−13^

## Data Availability

The data presented in this study are available on request from the corresponding author.

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
