# Peer review of "Optimization of a Monobromobimane (MBB) Derivatization and RP-HPLC-FLD Detection Method for Sulfur Species Measurement in Human Serum after Sulfur Inhalation Treatment"

_antioxidants, 2022, doi:10.3390/antiox11050939_

Round 1
Reviewer 1 Report
Under the perspective of this reviewer, the present manuscript, together with those from Shen et al. (Free Radical Biology and Medicine, 2012) and Ditrói et al. (Free Radical Biology and Medicine, 2019) share the discussion on the variables to take into account when applying the HPLC method for measurement of sulfide species in biological samples after MBB derivatization with fluorescence detection. These methodological details are of capital importance for the accurate measurement of these species and the interpretation of the obtained results.
Major points:
1) There are several parameters (such as temperature of the reaction, MBB concentration, free sulfide concentrations found in human serum samples, influence of O2, influence of light, among others) that are not unanimously chosen by the above mentioned papers. In this way, the manuscript from Roda et al. should raise these points, explain and discuss them thoroughly in comparison with the previously published papers.
2) Along the whole text, the term "hydrogen sulfide" - H2S (as being the actual chemical species measured in biological samples) is not strictly correct, as the "H2S-related species" present in liquid biological samples can be sulfide anions (S2-, HS-), H2S itself and the different forms of bound sulfane and acid-labile sulfur, which are well differentiated in the method according to the sample treatment (i.e., no treatment, DTPA or TCEP). In addition, it should be taken into account that the method of derivatization with MBB is run at pH 9.5, thus meaning that the sulfide anions HS- and S2- are the main constituents in the reaction mix (with just traces of real H2S).
3) When the authors refer to the mobile phase composition along the run, it is not clear from the written text that they are actually describing a gradient, but rather a sequence of isocratic compositions (changed by steps at 3, 16 and 20 min). This methodological aspect should be made more clear.
4) In order to understand which is the meaning of each of the parameters shown in Table 2, it is necessary to write the equation used for the standard curve fitting (in this case, a parabolic second-degree equation, y = A x2 + B x + C, where "y" represents AU and "x" represents sulfide concentration), as well as the definition of RMSE. These same considerations apply to Tables 3 and 4 (although in these cases the parameters belong to a linear equation y = A x + B), but the doubts regarding the meaning of each of the abbreviated parameters in the tables still remain.
5) The authors state the use of an UPLC system coupled to a QToF MS; however, in the legend to Figure 1, they wrote "MS/MS spectrum" where no reference is made to the chosen fragmentation transition and the detected daughter ion.
Minor point:
Figure 2 legend: correct "1.5 mM (red) e 0.15 mM (blue) MBB" to "1.5 mM (red) and 0.15 mM (blue) MBB".
Author Response
Reviewer: 1
The authors want to thank Reviewer 1 for having revised our original manuscript. All comments have been accepted and revision performed accordingly.
Comments:
Under the perspective of this reviewer, the present manuscript, together with those from Shen et al. (Free Radical Biology and Medicine, 2012) and Ditrói et al. (Free Radical Biology and Medicine, 2019) share the discussion on the variables to take into account when applying the HPLC method for measurement of sulfide species in biological samples after MBB derivatization with fluorescence detection. These methodological details are of capital importance for the accurate measurement of these species and the interpretation of the obtained results.
We appreciate your positive comments, highlighting constructive criticisms.
Suggestions: (Major points)
- There are several parameters (such as temperature of the reaction, MBB concentration, free sulfide concentrations found in human serum samples, influence of O2, influence of light, among others) that are not unanimously chosen by the above-mentioned papers. In this way, the manuscript from Roda et al. should raise these points, explain and discuss them thoroughly in comparison with the previously published papers.
Response: Thank you for your valuable remarks, which let us notice that the previous version of the manuscript was lacking a detailed analysis of methodological parameters influencing the analysis.
We agree that methodological details are extremely important; indeed, with the presented work we aimed to revise the conditions for the HPLC-FLD method for H2S speciation and quantification and provide a further optimization of some parameters, such as temperature for the derivatization reaction, samples handling (including dilution and tubes for speciation protocols) and sample aging conditions. The overall aim of the study was the definition of a standardized and robust protocol for the analysis which limits the interference on the actual H2S levels. Moreover, a calibration procedure was validated by a leave-one-out cross-validation (CV) for the first time.
We fully agree with the Reviewer and have largely revised the text of this revised version by adding paragraphs to the Introduction (line 94-120) and Discussion (623- 667).
In particular, we have highlighted all the issues comprehensively discussed in previous publications such as oxygen concentration and exposure to light, pH condition for derivatization reaction, trapping time on the speciation protocol. In the final protocol we maintained some of the experimental conditions previously published after having confirmed the absence of any significant difference (pH 9.5 Tris HCl as buffer, 30 min, 1% O2, dark environment). Finally, we included the parameters which this work aimed at optimizing such as MBB concentration, temperature for the derivatization reaction, and different sample handling conditions. The discussion of each technical parameter has now been updated and expanded in the revised version of the manuscript to include a comprehensive view of what previous authors has done and what aspects this work is improving and optimizing. Now we hope that the original aim of the study and our contribution to the state-of-the-art is clearer.
- Along the whole text, the term "hydrogen sulfide" - H2S (as being the actual chemical species measured in biological samples) is not strictly correct, as the "H2S-related species" present in liquid biological samples can be sulfide anions (S2-, HS-), H2S itself and the different forms of bound sulfane and acid-labile sulfur, which are well differentiated in the method according to the sample treatment (i.e., no treatment, DTPA or TCEP). In addition, it should be taken into account that the method of derivatization with MBB is run at pH 9.5, thus meaning that the sulfide anions HS- and S2- are the main constituents in the reaction mix (with just traces of real H2S).
Response: Thank you for valuable comment. We agree with the Reviewer that in the previous version of the manuscript we have been not accurate in the definition of the different H2S species when we mentioned them. In particular, the previous manuscript was lacking a paragraph discussing the concept that H2S can exist as different species in water. Throughout the manuscript, we have now specified the difference of H2S free, HS- and S2-, H2S pool, H2S species when relevant (lines 50-55 Introduction). Moreover, we specified H2S release from sulfide bond at specific pH and reducing condition. According to your suggestion, we revised several parts of the manuscript (Introduction lines 50-55, 60-67; Materials and methods lines 214-216, 233). However, we maintained the general term of H2S when referring to general biological-pathological features reported in papers which have not analyzed H2S species.
- When the authors refer to the mobile phase composition along the run, it is not clear from the written text that they are actually describing a gradient, but rather a sequence of isocratic compositions (changed by steps at 3, 16 and 20 min). This methodological aspect should be made clearer.
Response: Thank you for your comment. According to your suggestion, we corrected the description as follows: “The gradient elution started at 85:15 (v/v) and decreased to 68% water in 3 min. Then, it remains the gradient downward trend drop to 55% water for 13 min. Afterwards, the system was kept in isocratic elution mode for 1 min and then brought back to initial conditions 85:15 (v/v) in 3 min and left in this condition for 3 min to stabilize the pressure of the chromatographic system, thus completing the chromatographic separation in 23 min.”
- In order to understand which is the meaning of each of the parameters shown in Table 2, it is necessary to write the equation used for the standard curve fitting (in this case, a parabolic second-degree equation, y = Ax2 + Bx + C, where "y" represents AU and "x" represents sulfide concentration), as well as the definition of RMSE. These same considerations apply to Tables 3 and 4 (although in these cases the parameters belong to a linear equation y = Ax + B), but the doubts regarding the meaning of each of the abbreviated parameters in the tables still remain.
Response: Thank you for your suggestion. According to your suggestion, we added few lines for indicating the meaning of each parameter shown in table 3, 4 and 5 respectively, in the figure legends.
RMSE is defined as the mean of the squared differences between the experimental responses and the response values recalculated by the model.
The resulting model equation is in the form: ?=?+??+??^2.
(?_????????????=?+??_????????????): the intercept (a) and the slope (b)
(in this case in the form ?=?+??)
- The authors state the use of an UPLC system coupled to a Q-Tof/MS; however, in the legend to Figure 1, they wrote "MS/MS spectrum" where no reference is made to the chosen fragmentation transition and the detected daughter ion.
Response: thank you for your suggestion
We apologize for the mistake in the description part of Figure 1b within the original manuscript. We now have replaced “MS/MS spectrum” with “MS spectrum”. The experiment type conducted by UPLC-QTOF-MS is the function of MS mode where the collision energy was set as off state. According to your comment, we cited the work published by Shen et al. (X. Shen et al. Nitric Oxide 2014, 97-104) as a corresponding reference.
- Figure 2 legend: correct "1.5 mM (red) e 0.15 mM (blue) MBB" to "1.5 mM (red) and 0.15 mM (blue) MBB".
Response:
We have now edited the text accordingly to Your suggestion, thank you.

Reviewer 2 Report
The mBBr derivatization + HPLC method is widely used in H2S and polysulfides detection. Authors of this manuscript optimized some steps to make this method more proper for human serum samples. I think some results also provide useful tips for the application in other samples, such as microorganisms and mammalian cells. I recommend publication of this paper after proper modification.
- In Figure 1a, the authors used water as a blank control, also used mBB as another control. What is the difference between these two, why their HPLC spectra are quite different? especially between 0~10 min.
- In Figure 1b, the peak of SDB is MS1 or MS2? if it is MS2, the authors needs to give mass information of the fragments generated from SDB.
- In the introduction section. There are some new detection methods reported recently, such as GFP based method (doi: 10.1021/acs.analchem.8b04634.) and spectroscopy based method (doi: 10.1016/j.redox.2019.101179), which are worthy to mention.
Author Response
Reviewer: 2
The authors want to thank Reviewer 2 for the comments. Revisions have been accepted and details added to the text accordingly.
Comments:
The mBBr derivatization + HPLC method is widely used in H2S and polysulfides detection. Authors of this manuscript optimized some steps to make this method more proper for human serum samples. I think some results also provide useful tips for the application in other samples, such as microorganisms and mammalian cells. I recommend publication of this paper after proper modification.
Response: We appreciate your highly positive comments.
Suggestions:
- In Figure 1a, the authors used water as a blank control, also used MBB as another control. What is the difference between these two, why their HPLC spectra are quite different? especially between 0~10 min.
Response: Thank you for letting us understand that this paragraph was not sufficiently clear.
According to your suggestion, we have now edited the text indicating that the aim of these two controls is different. Standard solutions were prepared by dissolving Na2S in water and they were used to attribute the SDB peak in standards samples. Water control represents the blank sample, in which must not compare the SDB peak. MBB control was prepared and diluted by acetonitrile and injected in HPLC-FLD to indicate the retention tome of MBB peak since it’s used as excess so a residual amount it is also found in the chromatograms of SDB product of derivatization. A paragraph was added to the description of results shown in Figure 1a (lines 292-298).
- In Figure 1b, the peak of SDB is MS1 or MS2? If it is MS2, the authors need to give mass information of the fragments generated from SDB.
Response:
In the description part of Figure 1b, there is a descriptive mistake, we corrected and replaced “MS/MS spectrum” with “MS spectrum”. The peak of SDB is MS1. We cited the work published by Shen et al. (X. Shen et al. Nitric Oxide 2014, 97-104) as a reference.
- In the introduction section. There are some new detection methods reported recently, such as GFP based method (doi: 10.1021/acs.analchem.8b04634.) and spectroscopy-based method (doi: 10.1016/j.redox.2019.101179), which are worthy to mention.
Response: Thank you for this valuable comment giving us the opportunity to improve our manuscript upon addressing these novelties in the state-of-the-art.
According to your kind suggestion, we added the recently reported references to our revised manuscript and added few lines to briefly introduce these two detection methods in the Introduction (lines 87-93).
Reviewer 3 Report
This paper is on the optimization of a monobromobimane (MBB) derivatization and RP-HPLC-FLD detection method for sulfur species measurement in human serum after sulfur inhalation treatment. It is an important topic because it is clear that monobromobimane assay is not limited to solely measuring measuring free sulfide concentrations in blood serum or plasma but is affected by the total sulfide pool and the analytical conditions: alkylation time, light exposure, tight temperature control, the actual monobromobimane concentrations used, pH, and/or the presence or absence of chelators not to mention the tubing used for blood sampling (Nagy 2014, Ditroi 2019, Radermacher 2021).
The manuscript is in need of English corrections at times the statements are completely meaningless. In particular the materials and methods section needs not only improved English but needs revision with clear and defined methods and their rational.
Lines 77-80 “At the present, the HPLC-FLD method based on the MBB derivatization is one of the most widely employed methods, with wide applications ranging from in vitro to clinical studies [26], due to its high sensitivity, and selectivity with the low limit of detection and its high analytical throughput.”
If the above is so then why do we need this particular contribution?
The statement is false and the reference (Jia 2019: “H2S-based therapies for ischaemic stroke: opportunities and challenges”) is incorrect and in this context irrelevant, recommend deleting altogether.
Lines 567-569 repeat the same misleading statements as above needs to be corrected!
Lines 95-96 the authors refute the above statement: “Moreover, very few papers reported the application of the HPLC-FLD method for the H2S species quantification in complex biological matrices such as human serum [31,32].”
Furthermore the authors neglected Bogdándi 2018 which not only used human serum but also point to the fact that the sulfide levels detected using the monobromobimane method depend not only on the incubation time but also on the concentration of the alkylating agent.
Lines 321-322 “increasing reaction times from 30 min to 120 min” what was the rational for starting with reaction times of 30min, why not start with 10min? It has been shown that using the monobromobimane method the reaction completes in less than 10 min Bogdándi 2018 et al.
Lines 463-470 make no sense
Lines 603-607 The calibration procedure was based on the analysis of standard solutions at different H2S concentration (and not to solutions of purified SDB sample obtained from a single concentration of standard solution, as described in the literature).
Please elaborate on the above, what exactly is described in the literature and how does your calibration procedure improve what is reported? The statement as it stands makes no sense and please provide references!!!
Author Response
Reviewer: 3
The authors want to thank Reviewer 3 for having revised our original manuscript. We hope that the revision work performed following these comments have enhanced the quality of the revised paper. The Introduction and Discussion sections were modified integrating the discussion of previous presented results and highlighting the optimization performed. Also, the calibration procedure was integrated with further discussion of calibration procedure (a Table 2 with additional results was added) and compared to previously presented papers on HPLC-FLD MBB method. The reference suggested by the reviewer was added to the text and discussed.
Comments:
This paper is on the optimization of a monobromobimane (MBB) derivatization and RP-HPLC-FLD detection method for sulfur species measurement in human serum after sulfur inhalation treatment. It is an important topic because it is clear that monobromobimane assay is not limited to solely measuring free sulfide concentrations in blood serum or plasma but is affected by the total sulfide pool and the analytical conditions: alkylation time, light exposure, tight temperature control, the actual monobromobimane concentrations used, pH, and/or the presence or absence of chelators not to mention the tubing used for blood sampling (Nagy 2014, Ditroi 2019, Radermacher 2021).
The manuscript is in need of English corrections at times the statements are completely meaningless. In particular, the materials and methods section need not only improved English but needs revision with clear and defined methods and their rational.
Response:
We apologize for the lack of sufficient comprehension of several sentences. We thank the Reviewer for the comment. We revised the Material and methods section, and the main text accordingly. We hope the text has now improved.
Suggestions:
- Lines 77-80 “At the present, the HPLC-FLD method based on the MBB derivatization is one of the most widely employed methods, with wide applications ranging from in vitro to clinical studies [26], due to its high sensitivity, and selectivity with the low limit of detection and its high analytical throughput.” If the above is so then why do we need this particular contribution?
Response:
We thank referee for the comment for letting us realize that the novelty of our work was not sufficiently clear in the original manuscript. We have now modified the text to better describe the state of the art for HPLC-FLD method and highlight the contribution of our work.
In particular, we agree that the sentence in lines 77-80 was not accurate; we have now deleted any reference to ‘most widely employed methods, with wide applications ranging from in vitro to clinical studies’ and to ‘high analytical throughput’, throughout the paper and we deleted the wrong reference. Among the analytical methods presented in literature for H2S quantification, the HPLC-FLD with MBB derivatization deserves to be mentioned since its interesting features that make it suitable for the robust quantification of H2S levels. In this context, the main aim of the presented work was to revise the conditions for the HPLC-FLD method for H2S speciation and quantification and highlight our optimization of some parameters such as temperature for the derivatization reaction, samples handling (including dilution and tubes for speciation protocols) and sample aging conditions. The thread was to obtain a standardized and robust protocol for the analysis with limited interference on the actual H2S levels. In addition, a calibration procedure was validated for the first time by a leave-one-out cross-validation (CV). Finally, an application for the use of H2S levels to distinguish biological samples subjected to different treatments was successfully shown.
We changed the sentence (Introduction lines 94-96) and added a summary of the discussion already reported for the HPLC-FLD MBB method (Introduction lines 102-123; Discussion lines 632-667) to better highlight the contribution of our work.
- The statement is false and the reference (Jia 2019: “H2S-based therapies for ischaemic stroke: opportunities and challenges”) is incorrect and in this context irrelevant, recommend deleting altogether.
- Lines 567-569 repeat the same misleading statements as above needs to be corrected!
Response:
According to your suggestion, we corrected the improper statement and deleted the reference.
- Lines 95-96 the authors refute the above statement: “Moreover, very few papers reported the application of the HPLC-FLD method for the H2S species quantification in complex biological matrices such as human serum [31,32].”
Response:
We thank reviewer for the important comment which let us understand that this sentence was not sufficiently clear. This sentence was intended to specify that only a few authors have described the application of the HPLC-FLD-MBB method for the quantification of the different forms of H2S (free, acid-labile and bound sulfane) in serum samples. In fact, most of the work reported protocols for the free H2S quantification; only Shen et al developed a well-defined protocol to extract different levels of H2S from the biological pool.
The sentence has been rewritten to make it clearer and more accurate (Introduction lines 121-123).
- Furthermore, the authors neglected Bogdándi 2018 which not only used human serum but also point to the fact that the sulfide levels detected using the monobromobimane method depend not only on the incubation time but also on the concentration of the alkylating agent.
Response:
We thank reviewer for the important comment. We apologize for having not included Bogdándi 2018 within the references of the original paper. Now, we added the reference to the text and we commented his contribution in the Introduction (lines 102-123) and Discussion (lines 632-661); where we discussed the issues related to the MBB derivatization procedure. In particular we highlighted that ‘at high concentration MBB can cleave longer dialkyl polysulfide chains and extract H2S from these bound sulfane sulfur pools thus shifting speciation of sulfur species; MBB can liberate sulfide from sulfide pools when increasing reaction times above 7-10 min’.
- Lines 321-322 “increasing reaction times from 30 min to 120 min” what was the rational for starting with reaction times of 30min, why not start with 10min? It has been shown that using the monobromobimane method the reaction completes in less than 10 min Bogdándi 2018 et al.
Response: Thank you for your remarks. We have carefully read Bogdándi 2018 and are aware of their important evidence.
In our work we focalized on already published papers on MBB method using HPLC-FLD for the sulfide quantification (ref 22, 27, 31, 35, 36) in serum samples. In particularly, we implemented the conditions reported by Shen et al. also because they were the only authors describing a speciation protocol for the free sulfide, acid-labile and bound sulfane quantification in blood. About reaction time, Shen et al (ref 22, 27, 35) selected as the best reaction time 30 min as opposed to 15 min, given that ‘highest recovery percentage was achieved after 30 min of trapping headspace sulfide gas, establishing this timeframe as optimal’; Other authors have reported different studies; Tan et al (ref 31) selected as the best reaction time 120 min, given that ‘shorter reaction time especially in the range of 15–90 min can cause poor repeatability because of variability in the delays between the individual steps of the derivatization reaction’. Wintner et al (ref 36) defined 10 min as the reaction time sufficient for complete a reproducible conversion of 0.2–5 µM reactive sulfide to SDB, however the protocol used is quite different from Shen et al.
We verified the conditions for derivatization already reported by Shen et al and we further investigated some parameters (MBB concentration, Temperature, and sample handling conditions). However, we agree with referee that, as point out by Bogdándi 2018, we cannot exclude that our measurement of ‘free sulfide levels” at least partially measured a release from long polysulfides instead of direct alkylation of free sulfide, therefore we included a paragraph in the discussion (lines 640-645). Moreover, we agree with referee, the data at 120 min is also not interesting for the throughput of the method. We deleted this result, and we keep 30 min as reaction time. Overall, in the original manuscript we thought more appropriated to implement the conditions reported by Shen et al, we could take in consideration to test 10 min as reaction time in the future.
- Lines 463-470 make no sense
Response:
Thank you for the comment. We agree that this sentence was not clear enough, we apologize. The paragraph has been now rewritten as follows: Indeed, high standard deviations for the calculated concentrations were obtained (data not shown) when the twelve unknown samples (four samples collected at three different times of H2S inhalation) were interpolated, making all the calculated concentrations not significantly different from zero. These results revealed a poor prediction ability of the parabolic model. However, we noticed that the interpolated areas of the unknown samples were always in the range 5-15 AU. This means that the unknown samples are interpolated in the lowest concentration range of the curve, far from the centroid (that is around 600 AU). In general, for all regression models, the standard deviation calculated for a projected sample is lowest if the sample is close to the model centroid, while it increases, also dramatically, in the external portions. Therefore, the high standard deviations could be due to the non-optimal performance of the model in that response region.
- Lines 603-607 The calibration procedure was based on the analysis of standard solutions at different H2S concentration (and not to solutions of purified SDB sample obtained from a single concentration of standard solution, as described in the literature).
Please elaborate on the above, what exactly is described in the literature and how does your calibration procedure improve what is reported? The statement as it stands makes no sense and please provide references!!!
Response:
Thank you for letting us realize that this sentence was not clear enough and giving us the opportunity to improve our manuscript upon addressing this criticism.
We added details on calibration procedure to better explain the quantitative protocol developed and his validation. In this regard, it is important to notice that the majority of papers who described the HPLC-FLD methods for H2S detection in serum samples do not give many details on calibration procedure and calibration curve determination. Shen et al mention the use of dilution of purified SDB to calculate the calibration curve (ref 35). In the original manuscript we did not included some experimental set-up data involving calibration standards. Based on Tan et. al. (ref 31r.), which use ‘spiked’ serum samples for the calibration procedure, we compared ‘the addition of Na2S to serum sample’ to ‘Na2S standards. We verified that after ‘the addition of Na2S to serum sample’, a portion of H2S can be trapped by the serum matrix making the standard addition to serum matrix not reliable as calibration procedure. In particular, the SDB detection was 20% lower at 5 μM Na2S concentration and 50% lower at 25 μM Na2S concentration. In the revised version we edited the text, added Table 2 with results on this point and comments.
Data and comments were added to the Results (paragraph 3.2.2. Calibration curve: optimization and validation lines 434-447); and Discussion (lines 668-684)
We would like to accurately describe believe that our procedure of standard preparation and analysis (standard solution preparation > standard solution dilution > standard solution derivatization > HPLC-FLD separation > calibration curve and analysis) is more accurate than the ones previously published. We hope now this point could better emerge within the text. Moreover, we for the first time validated the calibration procedure, by using a a leave-one-out cross-validation (CV), as detailed in the original manuscript.

Round 2
Reviewer 1 Report
The authors have properly addressed all the points raised by the reviewers and included them in the new revised version of the manuscript.